# Investigation of Displacement and Ionospheric Disturbance during an Earthquake Using Single-Frequency PPP

**Jie Lv, Zhouzheng Gao *⬡, Cheng Yang ⬡, Yingying Wei and Junhuan Peng**

School of Land Science and Technology, China University of Geosciences Beijing, 29 Xueyuan Road, Beijing 100083, China
* Correspondence: zz.gao@cugb.edu.cn; Tel.: +86-186-2791-5172

**Abstract:** Currently, it is still challenging to detect earthquakes by using the measurements of Global Navigation Satellite System (GNSS), especially while only adopting single-frequency GNSS. To increase the accuracy of earthquake detection and warning, extra information and techniques are required that lead to high costs. Therefore, this work tries to find a low-cost method with high-accuracy performance. The contributions of our research are twofold: (1) an improved earthquake-displacement estimation approach by considering the relation between earthquake and ionospheric disturbance is presented. For this purpose, we propose an undifferenced uncombined Single-Frequency Precise Point Positioning (SF-PPP) approach, in which both the ionospheric delay of each observed satellite and receiver Differential Code Bias (DCB) are parameterized. When processing the 1 Hz GPS data collected during the 2013 Mw7.0 Lushan earthquake and the 2011 Mw9.0 Tohoku-Oki earthquake, the proposed SF-PPP method can provide coseismic deformation signals accurately. Compared to the results from GAMIT/TRACK, the accuracy of the proposed SF-PPP was not influenced by the common mode errors that exist in the GAMIT/TRACK solutions. (2) Vertical Total Electron Content (VTEC) anomalies before an earthquake are investigated by applying time-series analysis and spatial interpolation methods. Furthermore, on the long-term scale, it is revealed that significant positive/negative VTEC anomalies appeared around the earthquake epicenter on the day the earthquake occurred compared to about 4–5 days before the earthquake, whereas, on the short-term scale, positive/negative VTEC anomalies emerged several-hours before or after an earthquake.

**Keywords:** single-frequency precise point positioning (SF-PPP); vertical total electron content (VTEC); earthquake displacement; ionospheric disturbance

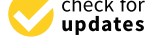



## 1. Introduction

Earthquakes will result in unexpected damages, especially in the regions around the epicenter. Some researches have been carried out to provide effective theoretical models to investigate the changes before, during, and after an earthquake [1–4]. Currently, technologies, for instance, the Interferometric Synthetic Aperture Radar (InSAR), seismometer, and Global Navigation Satellite Systems (GNSS) are usually utilized to inverse the displacements caused by an earthquake. These technologies have been proven to be effective in certain situations, but all of them have limitations. For example, it is challenging to use the InSAR-based method to obtain the deformation timely [5] owing to the long repetition period of InSAR satellites. Meanwhile, the solutions from a seismometer may drift due to the baseline errors and sensor errors [6]. Moreover, a broadband seismometer may clip large earthquake signals [7]. Even if strong-motion seismometers do not clip, the displacements transformed from acceleration instruments would decrease obviously because of the tilt and non-linear behaviors of accelerometers [8].

These drawbacks may be overcome by GNSS because of its high accuracy in positioning. GNSS has been used for the investigation of rupture processes [9], coseismic

displacement inversion [10], coseismic slip distribution [11], and preseismic deformation [12]. Compared to other technologies like the InSAR-based method [5] and the gravity recovery and climate ex-periment (GRACE)-based method [13], GNSS has advantages such as low-cost hardware, wide-coverage signal, high-accuracy observation, and the capability of all-weather monitoring. In addition, GNSS-measured displacements will not suffer from drift, clipping, or instrument tilting errors. These characteristics make GNSS particularly valuable in earthquake monitoring [4].

In general, two GNSS high accuracy positioning methods, namely, Post Processing Kinematic (PPK) and Precise Point Positioning (PPP) can be used in coseismic displacement inversion. The PPK method is used as an efficacious technique, because it adopts the double-differenced mode, which removes common biases such as the satellite orbit-related biases, atmosphere-related errors, and receiver-dependent offsets. There are extensive researches on the PPK-based GNSS seismology. For example, Xiang et al. [14] processed high-rate GNSS data during the Mw6.0 Italy earthquake in 2016 by using the TRACK module of GAMIT/GLOBK and compared its solutions with that of PPP. The results illustrated that the seismic signals have impacted the noise characteristic of GPS-derived displacement waveforms. Song et al. [2] inverted the coseismic deformation of the 2016 Kaikoura (New Zealand) earthquake by processing the 1 Hz GNSS observations using both the TRACK module and Principal Component Analysis (PCA) spatial filtering. The challenge for PPK-based approaches is that the reference station may also be displaced in strong earthquakes, e.g., the Mw9.0 Tohoku-Oki one [4], even when they are several hundred kilometers away from the epicenter. Moreover, the PPK positioning accuracy will be significantly degraded when the baseline length increases because of the residuals of atmospheric delays after applying the double-differenced model.

Compared to PPK, PPP [15] uses observations from a single station and precise satellite orbit/clock products afforded by the International GNSS Service (IGS) [16] centers. Most of the observation errors will be compensated by conventional models, while the corresponding residuals will be modeled as parameters. Thus, PPP can also reach high accuracy (i.e., centimeter or even millimeter level) after ambiguity convergence. Therefore, PPP has potential capability in displacement monitoring. For example, Li et al. [4] demonstrated the performance of the real-time ambiguity-fixed PPP using the 5 Hz Global Positioning System (GPS) data during the El Mayor-Cucapah earthquake (Mw7.2). Results illustrated that the PPP-based displacements are in accordance with the accelerometer-based ones at centimeter level. According to works in Geng et al. [17], two-thousand 24 h displacements at ninety-nine stations in Europe were analyzed by introducing GLONASS data into high-rate GPS PPP. Results reflected that the integration of GPS and GLONASS can further reduce the noise of high-rate PPP. Meanwhile, it also illustrates that the high-rate multi-GNSS PPP could be further amplified by using sidereal filtering. Tu et al. [3] studied the key sections of the integration between GNSS and strong motion records, which could be a guide for earthquake real-time monitoring and early warning.

Theoretically, there would be atmospheric disturbances before or after seismic activity [18–20]. However, researchers such as Rishbeth [21] and Nagao et al. [22] maintained that the atmospheric earthquake precursor is controversial and cannot be used to predict earthquakes [23]. Nevertheless, there are still many research works focusing on the inter-relationship between earthquake preparation and atmospheric disturbances (i.e., ionospheric disturbance). For example, a study by Leonard et al. [24] showed an anomalous disturbance of ionosphere electron density before the Alaska earthquake in 1964. Liu et al. [25] used GIM (Global Ionosphere Map) data to present the variations of Total Electron Content (TEC) during 35 earthquakes (Mw $\geq$ 6.0) that occurred from 1 May 1998 to 30 April 2008 in China. This method was also employed to analyze the seismo-ionospheric anomalies during the Wenchuan earthquake (Mw 7.9), China, on 12 May 2008. Tojiev et al. [26] analyzed the ionospheric anomalous phenomenon of strong earthquakes (Mw5.0) that occurred principally in and around the seismically active regions from 2006 to 2009 in Uzbekistan. The results showed that the ionospheric TEC may increase or decrease ab-

normally before or during earthquakes. Song et al. [27] found that two affairs of coseismic ionospheric disturbances were observed after the Wenchuan earthquake at 06:28 UT on 12 May 2008 through a GPS network in China. According to their conclusions, such two coseismic ionospheric disturbances were induced by the acoustic gravity waves that were active due to the partial transformation of the acoustic waves (because of the earthquake energy release). Tariq et al. [28] found that the seismic ionospheric anomalous phenomenon may occur within 10 days of an earthquake rupture by applying a temporal and spatial statistical analysis on the GPS- and GIM-TEC data from the three earthquakes (≥Mw7.0) near the Nepal-Iran-Iraq border in the course of 2015–2017. The reason for such positive anomalies in TEC might be owing to the existence of enormous energy in the epicenter before the earthquake ruptures. Ravanelli et al. [1] combined the ground displacement and ionospheric TEC in the Total Variometric Approach (TVA). This method was utilized in 2015 Illapel earthquake (Mw 8.3). The results showed that the ground motion and the TEC perturbation were detected at 30 s and 9.5 min after the rupture, respectively. The GNSS data stream by TVA opens new orientations to real-time warning systems for tsunami genesis estimation.

In general, those research works at present are chiefly based on dual-frequency GPS data which is a high cost. Thus, this paper aims to provide a method that can present the displacement and ionospheric disturbance simultaneously and economically. Compared to the previous works, the contribution of this paper could be itemized as: (1) it suggests an improved earthquake displacement estimation approach by considering the relationship between the earthquake and ionospheric disturbance. An undifferenced uncombined Single-Frequency PPP (SF-PPP) that parameterizes the ionospheric delay of each observed satellite is presented. (2) The relation between deformation and ionospheric disturbance is investigated by applying time-series analysis and spatial interpolation methods. For assessments, 1 Hz high-rate GPS data during the Lushan earthquake (Mw7.0, 2013) and Tohoku-Oki earthquake (Mw9.0, 2011) are processed and analyzed.

The paper is organized as follows: the Methodology section illustrates mathematical models of proposed methods SF-PPP and ionospheric calculation. The Results and Discussion section investigates the displacement and ionospheric disturbance obtained from the proposed method. This is followed by the Conclusions section.

## 2. Methodology

In this section, we describe the technical details of undifferenced uncombined SF-PPP and the Inverse Distance Weighting (IDW) method to invert the distribution of the Vertical Total Electron Content (VTEC).

### 2.1. Phase and Code Raw-Observation Functions

The observation functions for the raw pseudorange and carrier phase could be presented respectively as

$$
P_{r,j}^s(t) = \rho_r^s(t) + c \cdot \delta t_r(t) - o^s - c \cdot \delta t^s(t - \tau_r^s) + T_r^s(t) + I_{r,j}^s(t) + c(d_{r,j} + d_j^s) \\
+ dm_r^s + e_r^s
\tag{1}
$$

$$
L_{r,j}^s(t) = \rho_r^s(t) + c \cdot \delta t_r(t) - o^s - c \cdot \delta t^s(t - \tau_r^s) + T_r^s(t) - I_{r,j}^s(t) + \lambda_j N_{r,j}^s \\
+ \lambda_j(b_{r,j} - b_j^s) + \delta m_r^s + \varepsilon_r^s
\tag{2}
$$

$$
\rho_r^s(t) = \left\| (\vec{r}^s(t - \tau_r^s) + d\vec{r}^s(t - \tau_r^s)) - (\vec{r}_r(t) + d\vec{r}_r(t)) \right\|
\tag{3}
$$

$$
b_{r,j} = \phi_{r,j}(t_0) + \frac{c}{\lambda_j} \delta_{r,j}
\tag{4}
$$

$$
b_j^s = \phi_j^s(t_0) + \frac{c}{\lambda_j} \delta_j^s
\tag{5}
$$

where $P$ and $L$ refer to the raw observations of pseudorange and carrier phase; the indices $s$, $r$, and $j$ stand for the satellite, receiver, and signal frequency, respectively; $t - \tau_r^s$ and $t$ are the signal transmitting and receiving time; $c$ is the velocity of light in vacuum; $\rho$ indicates the geometric distance between the satellite and receiver; $o^s$ denotes the satellite orbit bias; $\delta t_r(t)$ and $\delta t^s(t - \tau_r^s)$ are the clock biases of the receiver and satellite; $T_r^s(t)$ and $I_{r,j}^s(t)$ are the tropospheric and ionospheric delay of the signal path at frequency $j$, respectively; $d_{r,j}$ and $d_j^s$ denote the hardware time delays of receiver and satellite on the pseudorange; $dm_r^s$ and $\delta m_r^s$ represent the multipath biases of pseudorange and carrier phase, respectively; $e_r^s$ and $\varepsilon_r^s$ represent the measurement noise of pseudorange and carrier phase; $\lambda_j$ and $N_{r,j}^s$ are the wavelength and integer ambiguity of carrier phase observations; $\overrightarrow{r}^s(t - \tau_r^s)$, $\overrightarrow{r}_r(t)$, $d\overrightarrow{r}^s(t - \tau_r^s)$, and $d\overrightarrow{r}_r(t)$ indicate satellite position, receiver position, antenna Phase Center Offset (PCO) [16] of satellite, and receiver PCO, respectively; $\phi_{r,j}(t_0)$ and $\phi_j^s(t_0)$ are the raw phase biases of receiver and satellite at $t_0$ epoch; $\delta_{r,j}$ and $\delta_j^s$ refer to the hardware time delays of receiver and satellite on the carrier phase; $b_{r,j}$ and $b_j^s$ are the comprehensive biases of receiver and satellite.

In addition, other errors that are not listed in the above equations, such as tidal loading, phase windup, relativistic delay, antenna Phase Center Variation (PCV) [16], etc., should also be corrected according to the conventional models [29–31].

### 2.2. Undifferenced Uncombined Single-Frequency PPP Model

The ionospheric delay is a key issue influencing the performance of SF-PPP. Therefore, several models, such as the GIM data correction model [32], the single-frequency GRAPHIC (GRoup And PHase Ionosphere Calibration) model [33], and the single-frequency ZIDE (Zenith Ionospheric Delay Estimation) PPP model [34], have been proposed. For example, Øvstedal [32] recommended using the GIM to correct the ionospheric delay on the pseudorange and carrier phase. The corresponding functions can be written as [35].

$$P_r^s = \rho_r^s + c \cdot dT_r + dTrop_r^s + dIon_r^s + \Delta_P \tag{6}$$

$$L_r^s = \rho_r^s + c \cdot dT_r + dTrop_r^s - \frac{c}{f_1} N_r^s - dIon_r^s + \Delta_L \tag{7}$$

where $dT_r$ is the receiver clock bias; $dTrop_r^s$ is the slant tropospheric delay; $dIon_r^s$ is the slant ionospheric delay which is obtained from the GIM model (Centre for Orbit Determination in Europe, CODE).

However, the drawbacks of the GIM-correction based SF-PPP model are that the residuals of each satellite's ionospheric delay and the un-disposed Differential Code Bias (DCB) of receiver will contaminate the performance of SF-PPP. To overcome these issues, Montenbruck [36] tried to use the GRAPHIC model [33] to eliminate the ionospheric delay on carrier phase instead of GIM data. In the GRAPHIC model, the characteristic that the ionospheric delays on pseudorange and carrier phase observations are equal in magnitude and opposite in sign for the same satellite is utilized. Hence, a linear ionospheric-free combination between pseudorange and carrier phase is adopted to form a new pseudorange plus carrier phase ionospheric-free combination [35].

$$P_r^s = \rho_r^s + cdT_r + dTrop_r^s + dIon_r^s + \Delta_P \tag{8}$$

$$G_r^s = \frac{(P_r^s + L_r^s)}{2} \tag{9}$$

where $G_r^s$ represents the linear ionospheric-free combination between pseudorange and carrier phase; the other symbols are the same as those described above. The noise of combined observation is much larger than the carrier phase observation because of the mixed pseudorange noise.

Even so, the SF-PPP positioning accuracy of GRAPHIC still cannot satisfy the accuracy requirement because of the large noise of carrier phase in the GRAPHIC model [37].

Therefore, based on the GRAPHIC model, Beran et al. [34,38] proposed the ZIDE model to estimate the zenith ionospheric delay of each satellite as a parameter, which has been verified to promote the performance of SF-PPP. The corresponding functions could be described as

$$P_1^k(t) = \rho^k(t) + c \cdot \delta t_{r,k}(t) - o^k - c \cdot \delta t^k(t - \tau_r^k) + T_r^k(t) + I_{r,1}^k(t) + c(d_{r,1} + d_1^k) \\ + dm_r^k + e_{P,1}$$ (10)

$$L_1^k(t) = \rho^k(t) + c \cdot \delta t_{r,k}(t) - o^k - c \cdot \delta t^k(t - \tau_r^k) + T_r^k(t) - I_{r,1}^k(t) + \lambda_1 N_{r,1}^k \\ + \lambda_1(b_{r,1} - b_1^k) + \delta m_r^k + \varepsilon_{L,1}$$ (11)

where the symbols are the same as those defined in Equations (1) and (2).

According to the works of Shi et al. [35], Le et al. [39] and Zhang et al. [40], ZIDE SF-PPP could improve position accuracy and convergence time by adopting precise global and regional ionospheric models. However, the receiver DCB on pseudorange is ignored in the ZIDE model. In this paper, the SF-PPP model on the basis of Ionospheric delay and Receiver DCB Constraint (IRC) [41] is applied to acquire the epoch-by-epoch time series of positions and VTEC values. Here, to separate the receiver DCB from ionospheric delay [42], the GIM data are utilized to generate a virtual external observation for each ionospheric delay. The basic equations could be presented as

$$I_{r,f_1}^s = 40.28 \cdot \text{VTEC} / \left( f_1^2 \cos(Z_\theta) \right) + \varepsilon_{I_{r,f_1}^s}, \varepsilon_{I_{r,f_1}^s} \sim N\left( 0, \sigma_{\varepsilon_{I_{r,f_1}^s}}^2 \right)$$ (12)

$$\begin{bmatrix} d_{DCB} \\ 0 \end{bmatrix} = \begin{bmatrix} 1 & -1 \\ \frac{f_1^2}{f_1^2 - f_2^2} & -\frac{f_2^2}{f_1^2 - f_2^2} \end{bmatrix} \begin{bmatrix} d_1 \\ d_2 \end{bmatrix}$$ (13)

where VTEC stands for the vertical total electron content acquired from the GIM model, and $Z_\theta$ is the zenith angle at the Ionospheric Punctuation Point (IPP); $\varepsilon_{I_{r,f_1}^s}$ is the accuracy of the GIM model with the prior variance of $\sigma_{\varepsilon_{I_{r,f_1}^s}}^2$; $d_{DCB}$ is the DCB between pseudoranges $P_1$ and $P_2$.

Then, the final parameter vector of SF-PPP can be written as

$$X = \left[ \mathbf{x}, c\delta t_r, \text{ZWD}_r, d_{DCB}, \overline{N}_r^s, I_{r,1}^s \right]^T$$ (14)

where $\mathbf{x}$ is the vector of the receiver position increments related to the prior position; $\text{ZWD}_r$ is the residual of wet components of the tropospheric delay; $\overline{N}_r^s$ is the reparameterized ambiguities; the other symbols are the same as those described above.

### 2.3. Parameters Adjustment

To estimate the parameter vector, sequential least square estimation [43] is utilized. According to the mathematical model in [44], we have the error function of

$$V_k = A_k \hat{X}_k - L_k, \ P_k = \sum_k^{-1}$$ (15)

$$V_{k-1} = A_{k-1} \hat{X}_{k-1} - L_{k-1}, \ Q_{k-1} = \sum_{\hat{X}_{k-1}}$$ (16)

where $V$ is the residual vector; $L$ is the innovation vector; $\hat{X}$ denotes the parameter vector; $A$ is the coefficient matrix; $P$ and $Q$ are the weight matrix and covariance matrix, respectively.

Based on parameters estimation algorithm, the following extremum criterion function is applied

$$\Omega = V_k^T P_k V_k + \left( \hat{X}_k - \hat{X}_{k-1} \right)^T P_{\hat{X}_{k-1}} \left( \hat{X}_k - \hat{X}_{k-1} \right) = \min$$ (17)

where $P_{\hat{X}_{k-1}} = \sum_{\hat{X}_{k-1}}^{-1}$.

Therefore, the state vector can be written as

$$\hat{X}_k = P_{\hat{X}_k}^{-1}\left(A_k^T P_k L_k + P_{\hat{X}_{k-1}}\hat{X}_{k-1}\right) \tag{18}$$

$$P_{\hat{X}_k} = A_k^T P_k A_k + P_{\hat{X}_{k-1}} \tag{19}$$

where the symbols are the same as those described above.

*2.4. SF-PPP Based Regional VTEC Model*

The ionosphere is an ionized atmosphere with a height of approximately 60 km to 1000 km. To model the TEC as mathematical function, the free electrons are usually assumed to be distributed in a thin shell at a median height of 450 km [45]. The conventional method to obtain TEC from dual-frequency GPS observations is the polynomial model [46], spherical harmonic model [47], polyhedral function model [48], etc. In recent years, the Ionospheric-Constrained PPP (IC-PPP) [42] realizes the estimation of ionospheric delay on the slant signal path of each observed satellite. According to the VTEC solution from the IRC SF-PPP in this paper, the IDW method [49] can be utilized to interpolate the VTEC at the given latitude and longitude of IPP.

The IDW method is a Shepard method [49] that provides weight for distant points by using the inverse of the distance or using a more elaborate function. Each observation is assigned to the interpolated point after weighting by the inverse of the distance. Figure 1 shows the main idea of the IDW method [49]. In the following, the index *i* represents an included IPP point; the index *j* is an interpolated point, and $n_j$ is the amount of included IPP points that correlate to the interpolated point.

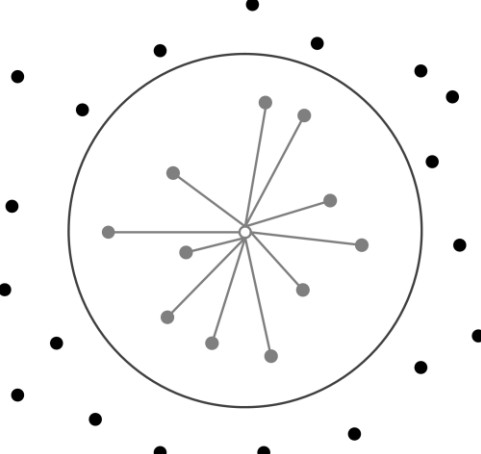

**Figure 1.** IPP points within a given radius; black points represent excluded IPP points; gray points dedicate included IPP points; white point is the interpolated point; the bold circle, which is centered on the interpolated point, separates excluded and included IPP points.

In the IDW method, the weighted mean of attribute values of the discrete points is adopted to estimate the attribute value of the included IPP point [49]. Therefore, how to get a reliable weight is the key to assess the quality of interpolation results. The $\overline{VTEC_j}$ at the interpolated point *j* could be described as

$$\overline{VTEC_j} = \frac{\sum\limits_{i=1}^{n_j} \frac{VTEC_{i,j}}{d_{i,j}^2}}{\sum\limits_{i=1}^{n_j} \frac{1}{d_{i,j}^2}} \tag{20}$$

where $VTEC_{i,j}$ is weighted by the inverse of distance given the average VTEC at the interpolated point; $d_{i,j}$ is the distance between the point $i$ and the interpolated point $j$. The IDW method usually sets the search distance within a sphere, and evaluates the estimated points by known points within the range. The weight of a known point is only determined by the distance between the known point and the one to be estimated, regardless of absolute position. This approach is convenient to implement, and the complexity in time and space is much smaller than many other methods (e.g., ordinary Kriging, co-Kriging, and empirical Bayes Kriging) [50].

## 3. Results and Discussion

To validate the performance of the proposed method in terms of dynamic deformation, ionospheric disturbance, and the inherent relation between earthquake and ionospheric disturbance, GPS observations from two earthquakes (the Mw7.0 Lushan earthquake on 20 April 2013 and the Mw9.0 Tohoku-Oki earthquake on 11 March 2011) are processed.

### 3.1. Data Processing Method

The 1 Hz GPS data provided by Japan's GEONET (GNSS Earth Observation Network System) and the Crustal Movement Observation Network Of China (CMONOC) are used for the analysis of the earthquakes in Tohoku-Oki and Lushan, respectively. During the data processing, the precise orbit and clock products from IGS are adopted. Those GPS data are processed under the modes of SF-PPP and GAMIT/TRACK.

### 3.2. Earthquake Displacement

As is known, the most direct expression of earthquakes is the induced surface deformation. Therefore, the surface deformations caused by those two earthquakes are analyzed concisely.

#### 3.2.1. Deformation of Tohoku-Oki Earthquake

The Mw9.0 Tohoku-Oki earthquake occurred on 11 March 2011, at 05:46:24 UTC in the east of the Oshika Peninsula of Tohoku is located at 38.10°N, 142.60°E, with a 30 km depth of epicenter. It is one of the best-recorded earthquakes in history since Japan built one of the densest GPS networks in the world. It is also the fourth-largest earthquake in the world since 1900. Table 1 lists the information of the selected stations and the distance between the epicenter and each station calculated by 1 Hz GPS data using GAMIT/TRACK and the proposed SF-PPP method. The distribution of the epicenter and the selected stations are presented in Figure 2. According to the works in [2], 32 GPS stations and station 0848 (825.2 km far away from the epicenter) are utilized as the rover station and base station, respectively.

**Table 1.** Epicentral distance of the selected GPS stations.

| Site | Distance (km) | Site | Distance (km) | Site | Distance (km) | Site | Distance (km) |
|------|---------------|------|---------------|------|---------------|------|---------------|
| 0906 | 191.9 | 0201 | 173.0 | 0156 | 284.1 | 0026 | 358.5 |
| 0167 | 160.8 | 0547 | 184.4 | 0543 | 244.8 | 0921 | 333.3 |
| 0172 | 126.3 | 0910 | 168.6 | 0553 | 240.5 | 0030 | 320.5 |
| 0918 | 122.1 | 0029 | 165.4 | 0926 | 233.9 | 0925 | 272.0 |
| 0549 | 126.6 | 0174 | 172.7 | 0190 | 223.6 | 0191 | 264.5 |
| 0550 | 98.8 | 0548 | 161.0 | 0554 | 219.2 | 0932 | 264.1 |
| 0037 | 146.1 | 0934 | 193.5 | 0033 | 212.1 | 0231 | 295.5 |
| 0919 | 145.4 | 0180 | 189.7 | 0199 | 221.3 | 0049 | 271.2 |

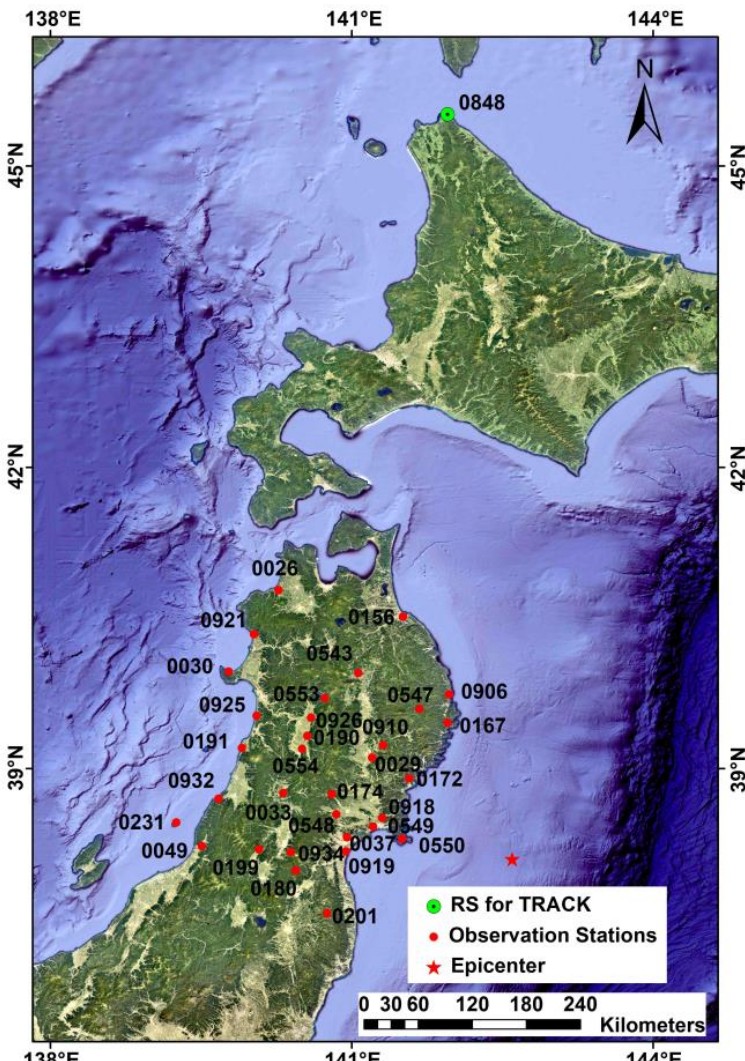

**Figure 2.** Distribution of the selected GPS stations and the location of the 2011 Tohoku-Oki earthquake epicenter; the green point represents the reference station.

As the earthquake occurred at 05:46:24 UTC, the dynamic displacements from 05:40:21 to 05:55:20 UTC were adopted to reflect the whole process. The results of eight rover stations were adopted as examples to illustrate the performance of the two methods. Meanwhile, since there was a Common Mode Error (CME) in all stations' sequences [51], it covered the weak or transient signals that had a visible influence on extracting the deformation characteristic of each station in the area network. Works from Gruszczynski et al. [52] suggest that CME might be engendered by satellite orbit biases, ocean loading tidal correction residuals, atmospheric tides ($S_1$ and $S_2$ tidal wave), etc., and it may be affected by the implementation path of the reference frame. In addition, while the earthquake impact range is wide, or the reference station is near to the rover station, the disturbance of the earthquake on the reference station also will be superposed on each observation station under relative positioning mode. This circumstance is named the coseismic interference of the reference station and will present itself as CME [2].

Figures 3 and 4 show the coseismic displacements of the GPS stations. Those stations move towards the southeast in the horizontal component. Both the GAMIT/TRACK and SF-PPP solutions can present accurately the displacement trend. However, the magnitude of the displacement, especially in the east, of the GAMIT/TRACK solutions is slightly (i.e., about 36.8 cm for 0172) different from the SF-PPP method. The reason is that the intensity scale of the Tohoku-Oki earthquake is strong enough, and the range of influence is

wide. It leads to large fluctuations in the reference station 0848 and results in displacement superposition. Moreover, the time series of GAMIT/TRACK in the east direction has an obvious CME. The regular method is eliminating the influence by using PCA spatial filtering. Nevertheless, as an absolute positioning method, the SF-PPP method can prevent the sequences from the influence of CME, which makes SF-PPP a potential low-cost technique in earthquake deformation inversion.

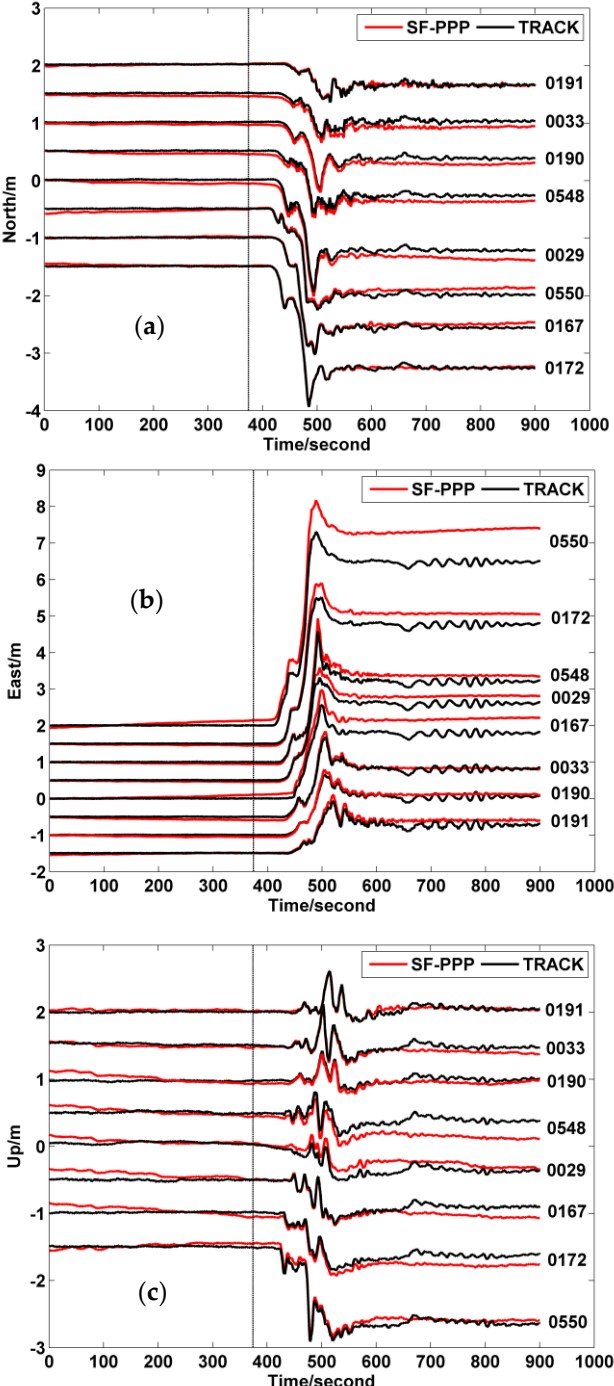

**Figure 3.** Time series for the three directions of GPS stations after 05:40:21 (UTC time), where in (**a**–**c**) are the position time series in the north, east, and up components.

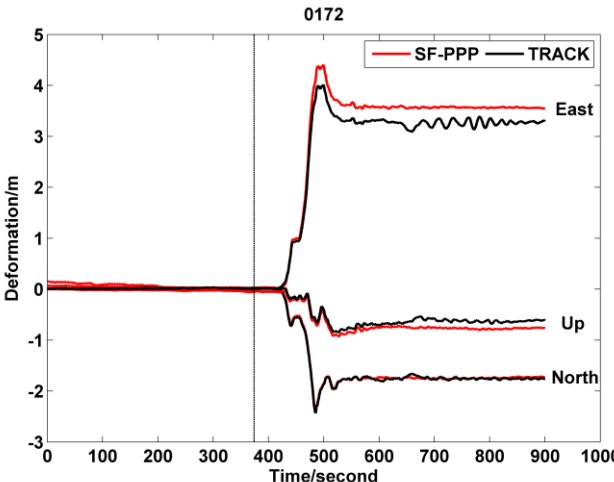

**Figure 4.** Time series for the three directions of 0172 station after 05:40:21 (UTC time).

The RMSs of the GAMIT/TRACK and SF-PPP position errors are enumerated in Table 2. After removing the solutions under poor satellite-observing conditions and during the convergence period, the position errors are calculated by comparing the solutions from the mean value between 0 and 5 h before the earthquake. Accordingly, it can be found that the accuracy of GAMIT/TRACK in terms of RMS is much higher than that of SF-PPP. Meanwhile, GAMIT/TRACK captures lower amplitude changes than that of SF-PPP because of its higher positioning accuracy character. Nevertheless, the results afforded by Figures 3 and 4 illustrate that SF-PPP provides the coseismic displacement with similar accuracy as that from GAMIT/TRACK. It is because the detection of coseismic displacement just requires high accuracy position solutions in the relative term. In addition, such an SF-PPP model also can provide ionospheric VTEC distribution. Hence, we can adopt SF-PPP to calculate and analyze the VTEC variation during the earthquake. Moreover, for both methods, the precision in horizontal directions is much higher than that in the vertical direction. This circumstance is engendered by the spatial distribution of GPS satellite constellations.

**Table 2.** RMS of the stations in the three directions in the methods of GAMIT/TRACK and SF-PPP.

| Site | RMS (cm) | | | | | |
|---|---|---|---|---|---|---|
| | North | | East | | Up | |
| | TRACK | SF PPP | TRACK | SF PPP | TRACK | SF PPP |
| 0029 | 2.67 | 48.88 | 1.14 | 23.48 | 3.98 | 58.97 |
| 0033 | 3.12 | 50.88 | 0.87 | 18.01 | 3.90 | 53.69 |
| 0167 | 1.95 | 57.61 | 1.03 | 45.54 | 4.46 | 148.32 |
| 0172 | 2.17 | 48.49 | 1.34 | 14.56 | 3.92 | 36.76 |
| 0190 | 2.66 | 58.33 | 1.02 | 20.60 | 3.93 | 66.42 |
| 0191 | 2.79 | 84.53 | 1.54 | 53.93 | 3.75 | 167.20 |
| 0548 | 2.84 | 47.91 | 1.02 | 18.59 | 4.46 | 51.62 |
| 0550 | 2.45 | 45.93 | 1.23 | 54.15 | 4.06 | 131.09 |

### 3.2.2. Dynamic Deformation of Lushan Earthquake

The Mw7.0 Lushan earthquake, which occurred at 00:02:46 on 20 April 2013, caused significant ground motions. Even about 5 cm deformations could be detected at distances of 42 km from the epicenter. Lushan county is located at the front edge of the southern section of the Longmenshan fault zone, which is part of the eastern edge of the Tibetan Plateau, where earthquakes occur frequently. In this research, 1 Hz GPS data of continuous sites of CMONOC during the 2013 Lushan earthquake are adopted to calculate the coseismic deformation and ionospheric anomalous distribution. Eight CMONOC stations near the

epicenter (as listed in Table 3) are chosen to analyze the surface deformation. The detailed information of the selected GPS stations is shown in Table 3. The corresponding geospatial distribution of those selected stations and the spatial geometry between the epicenter and those selected stations are presented in Figure 5.

**Table 3.** Epicentral distance of the selected GPS near field stations.

| Site | Distance (km) | Site | Distance (km) |
|------|---------------|------|---------------|
| SCDF | 214.4 | SCSM | 134.8 |
| SCJL | 210.0 | SCSN | 231.4 |
| SCMB | 155.6 | SCTQ | 42.7 |
| SCMX | 174.2 | SCXJ | 116.8 |

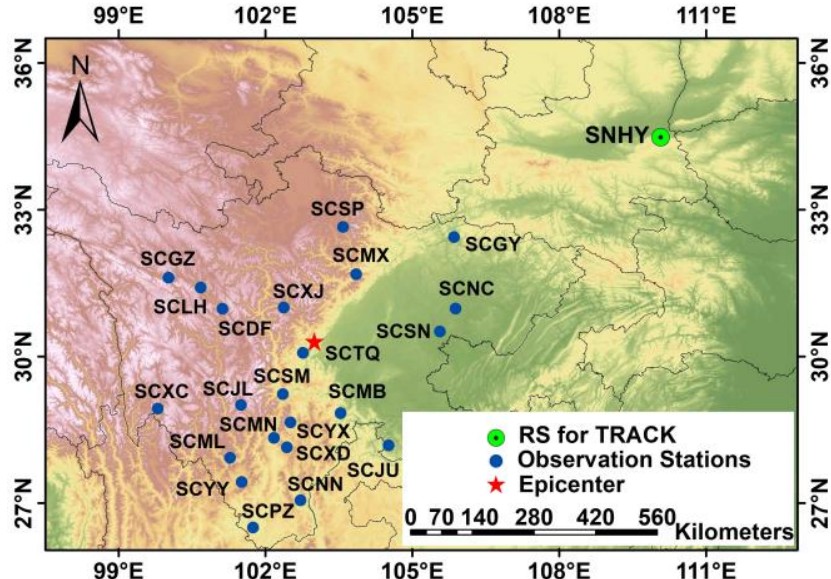

**Figure 5.** Location of the 2013 Lushan earthquake epicenter and the corresponding distribution of continuous stations of CMONOC in Sichuan Province; the green point represents the reference station used in GAMIT/TRACK.

Here, we provide the dynamic displacement of the first 10 min on 20 April (UTC time) that are calculated by GPS data before, during, and after the earthquake. The solutions calculated by GAMIT/TRACK are plotted in Figure 6, from which we can see that the deformation engendered by the Lushan earthquake is significant at only the SCTQ station but rather slight at the other stations. The reason is that SCTQ is the nearest station to the earthquake epicenter with a distance of about 42 km, while the other stations are more than 110 km even up to 230 km away, which can be found in Table 3 and Figure 5. The deformations at the SCTQ station, which are calculated by GAMIT/TRACK and SF-PPP, are shown in detail in Figure 7.

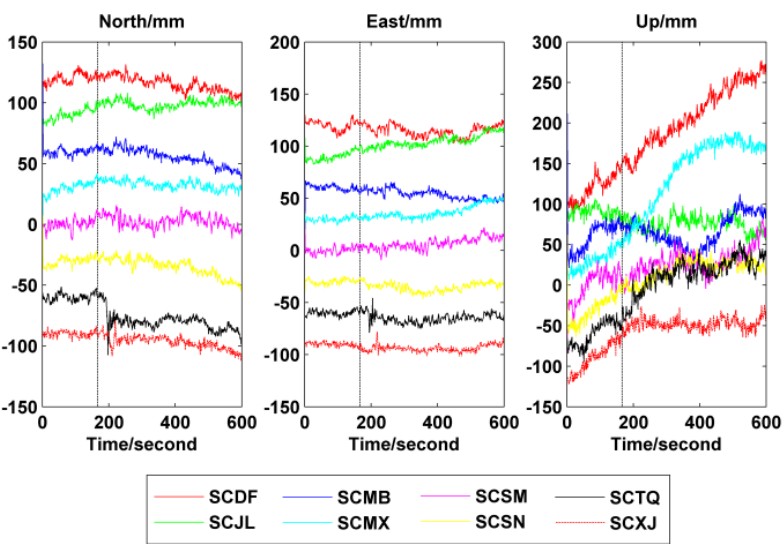

**Figure 6.** Time series for the three directions of selected stations from GAMIT/TRACK after 00:00:00 (UTC time); the black dotted line stands for the time of earthquake occurred.

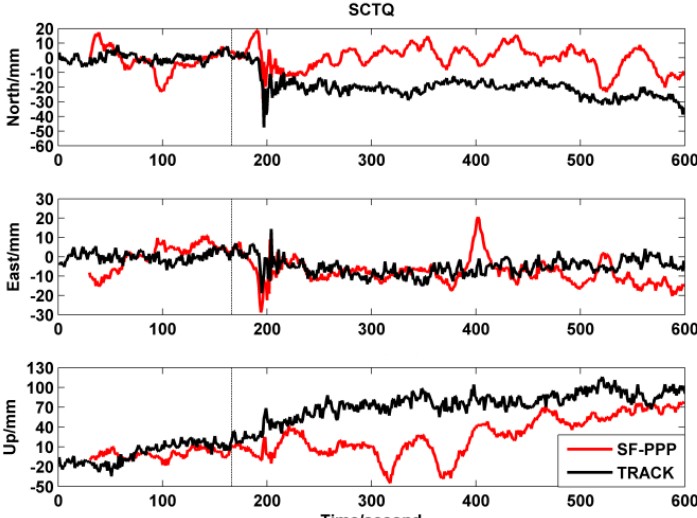

**Figure 7.** Time series for the three directions of SCTQ station after 00:00:00 (UTC time), respectively; the black dotted line stands for the time of earthquake occurred.

To evaluate the accuracy of our method, the solutions from GNSS data products of the China Earthquake Administration (http://www.cgps.ac.cn, accessed on 27 August 2019) are adopted as the reference. According to the comparison results, it can be known that both GAMIT/TRACK and SF-PPP can record the coseismic deformation, with the deformation of about 4–5 cm in the north and about 2–3 cm in the east. However, as shown in Figure 7, the solutions from GAMIT/TRACK are more stable than those of SF-PPP. It may be because the solution of such SF-PPP model depends highly on the accuracy of virtual observation of ionospheric delay, the accuracy of pseudorange, and the estimation strength of ambiguities and ionospheric delay. The measurement accuracy of pseudorange is empirically about 0.3 m. Since the GIM data are utilized as virtual observation of ionospheric delay, and the accuracy for GIM data is about 2–8 TECU [53], the accuracy of the ionospheric delay virtual observation is around 0.33–1.32 m errors.

Compared with the conventional SF-PPP, the proposed SF-PPP model has more parameters, which may weaken the positioning solutions. On the other hand, as is proven by Gao et al. [41], such a proposed SF-PPP model can reduce the position solutions noise obviously. Thus, it is possible to use its position solutions to identify the surface deformation during

the earthquake. What is more, the proposed SF-PPP model can also provide the ionospheric VTEC distribution.

Figure 6 depicts the dynamic deformation for the three directions in the first 10 min of the eight selected stations. It can be seen that deformations of SCTQ, SCSM, and SCXJ are relatively evident, especially in the horizontal direction. Meanwhile, the deformation amplitude values of those stations degrade clearly along with the distance between epicenter and track stations. For instance, the deformation values for the SCTQ station in the north and east components are 47.4 mm and 25.4 mm, respectively. By comparison, the deformations of SCSM and SCXJ are less significant, especially in the vertical direction. For the SCMB station, only the signal in the east direction can be identified, while signals in the north and up directions are submerged by noises. However, although the SCSN station is located far from the earthquake area, it has an obvious change in the amplitude and frequency when the seismic wave approaches. This phenomenon is due to the fact that seismic signals far from the epicenter are difficult to identify directly from the time series. In addition, because the positioning accuracy of the GNSS in vertical is worse than that in horizontal, the seismic signals in the vertical component are submerged by the noise, which makes it difficult to detect. Generally, since the Lushan earthquake belongs to the thrust earthquake with the character of levorotation, the mainshock occurred on the thrust fault with a large dip angle, and the scale of earthquake rupture was minor.

### 3.3. Ionospheric Disturbance during Earthquake

Besides the displacement, the earthquake also would cause ionospheric disturbances. Therefore, the VTEC variation during the earthquake is analyzed in this section. The ionospheric VTEC from SF-PPP will be used to interpolate the VTEC above the epicenter. Thus, the accuracy of VTEC interpolation highly relies on the geographical distribution of GPS observation stations around the earthquake location. According to Figure 2, the GPS stations utilized in the Tohoku-Oki earthquake present an abnormal geographical distribution. Therefore, only the ionospheric disturbance in the Lushan earthquake is analyzed in detail.

Generally, the direct phenomenon of the seismic-ionospheric coupling is atmospheric oscillation. When an earthquake happens, gravity waves will be generated, and propagated to the ionosphere, causing ionospheric anomaly disturbances [54]. While pursuing the origin of atmospheric oscillation excitation, there are three possibilities [55]: (1) the block structure produces an undulating motion during the earthquake preparation; (2) the heating of greenhouse gas released near the active fault causes abnormal thermal radiation in the near-surface atmosphere [56]; (3) the inert gas is released near the active fault [57]. Moreover, according to the works of Hines [20], Occhipinti [19], and Astafyeva [18], earthquake acoustic waves can also cause perturbations in the ionosphere. The earthquake energy is released in the shape of seismic waves.

We adopt GPS data from 22 CMONOC stations in Sichuan Province to investigate ionospheric anomalies during the Lushan earthquake. The distribution of those stations is shown in Figure 5. Solar and geomagnetic activities and variations in weather bring a strong disturbance effect on the ionosphere. Therefore, it is necessary to eliminate these effects before identifying whether an ionospheric anomaly is engendered by the earthquake. According to Jiang et al. [58], during the Days Of Year (DOY) of the 90–120 period, solar activity was at a low level (http://www.sepc.ac.cn/, accessed on 27 August 2019) and weather conditions were normal, except for the DOY 101, 102, and 112. The geomagnetic indices were extracted on the website http://wds.kugi.kyoto-u.ac.jp/, accessed on 27 August 2019. According to Dst and Kp, from April 13th to 27th (DOY 103–117), there were geomagnetic disturbances on the DOY 114–116, and the geomagnetic field was low on the other days.

The IDW method mentioned in the SF-PPP-based regional VTEC model part is utilized to analyze the distribution of VTEC near the epicenter before and after the earthquake. The VTEC data on DOY 105, 106, 109, and 110 (the day earthquake happened) are applied for

spatial interpolation. The seismic-ionosphere effect and the spatial distribution/variation of ionospheric anomaly calculated by SF-PPP are presented in Figure 8. In addition, the time series of VTEC on DOYs 105, 106, and 110 are shown in Figure 9. It is obvious that the VTEC on DOY 110 is lower than DOYs 105 and 106. In contrast, the corresponding solutions from GIM data are shown in Figure 10. According to Figure 8, compared to DOY 105 and 106 before the earthquake, the VTEC near the earthquake epicenter decreased on DOY 110. However, the location of the VTEC anomaly area was inconsistent with the vertical projection of the earthquake epicenter. Such an area appeared significantly in the south of the epicenter, while the VTEC was relatively lower above the earthquake zone. Meanwhile, there were no reports of obvious or approximate VTEC anomalies in other parts of the whole world before and after the Lushan earthquake [58]. These pieces of evidence further prove that the ionospheric anomaly was associated with the Lushan earthquake.

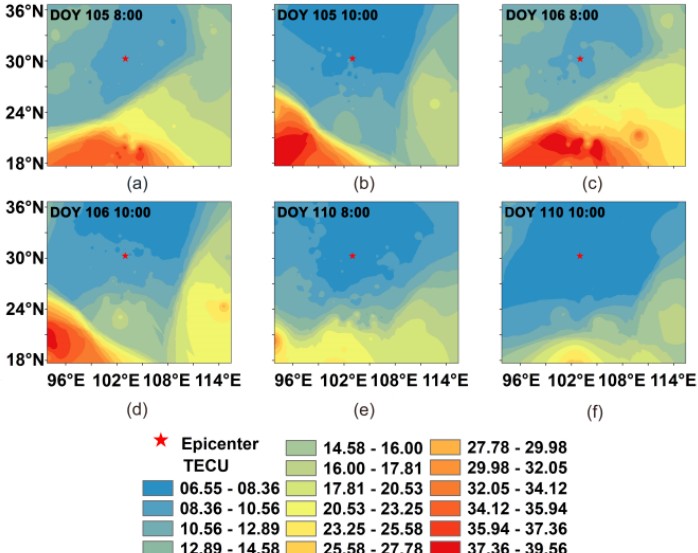

**Figure 8.** Distribution of VTEC near the earthquake epicenter on DOY 105, 106, and 110, 2013 (the day earthquake happened); (**a**,**b**) indicate distribution of VTEC at 8:00 and 10:00 of DOY 105; (**c**,**d**) stand for distribution of VTEC at 8:00 and 10:00 of DOY 106; (**e**,**f**) denote distribution of VTEC at 8:00 and 10:00 of DOY 110; the time here indicates UTC time.

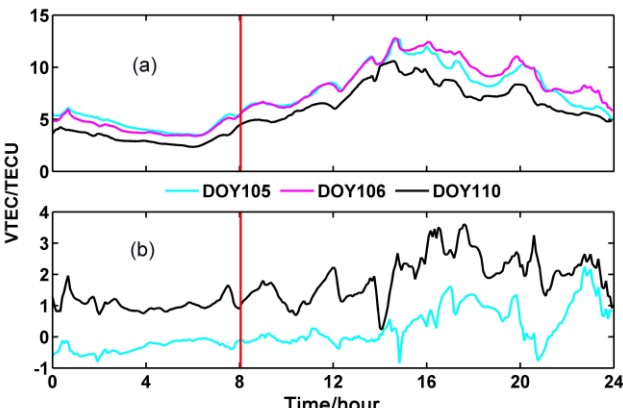

**Figure 9.** VTEC time series of the earthquake epicenter on DOY 105, 106, and 110, 2013 (the day earthquake happened): (**a**) is the VTEC time series; (**b**) is the difference between VTEC on DOY 106 and VTEC on DOY 105 and 110, 2013; the red line stands for the time of earthquake occurred; the time here indicates Beijing time.

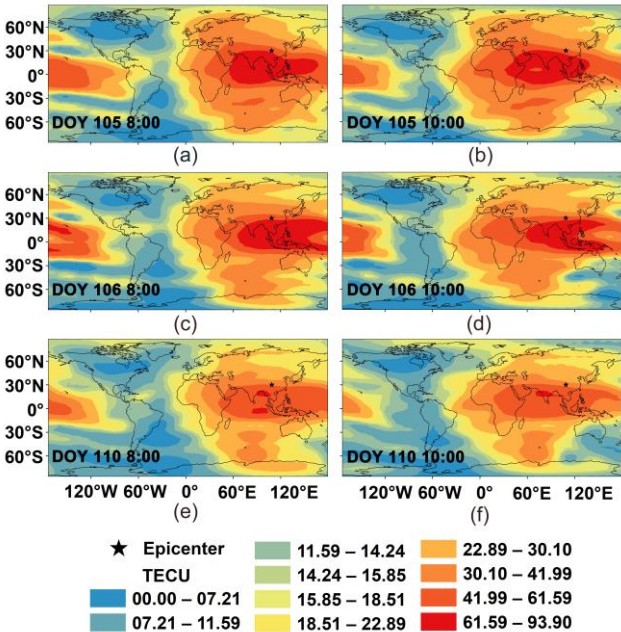

**Figure 10.** Distribution of VTEC of the world on DOY 105, 106, and 110, 2013 (the day earthquake happened) from GIM data; (**a**,**b**) indicate distribution of VTEC at 8:00 and 10:00 of DOY 105; (**c**,**d**) stand for distribution of VTEC at 8:00 and 10:00 of DOY 106; (**e**,**f**) denote distribution of VTEC at 8:00 and 10:00 of DOY 110; the time here indicates UTC time.

According to the results in Figure 11, further research on the ionospheric anomaly on DOY 109 and 110 is presented. Here, we mainly focus on the variation of VTEC during the periods of several hours before and after the earthquake. The IDW method mentioned in the SF-PPP-based regional VTEC model part is adopted to estimate the distribution of VTEC near the epicenter before and after the earthquake. The VTEC data on DOY 109 and 110 were applied for spatial interpolation. The corresponding solutions are portrayed in Figure 11, from which we can know that the VTEC decreased firstly and then increased gradually. Moreover, the feature of the VTEC distribution was similar to the results in Figure 8. Thus, the location showed a higher VTEC distributed in the south area of the epicenter. Figure 11d also shows the VTEC distribution near the earthquake rupture. After the earthquake occurred, the VTEC started to raise obviously.

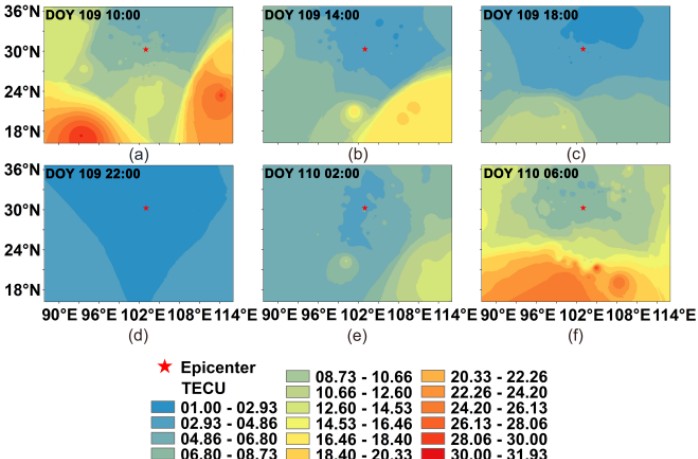

**Figure 11.** Distribution of VTEC near the earthquake epicenter from 10:00 UTC on DOY 109 to 6:00 UTC on DOY 110, 2013 (the day earthquake happened); subfigures (**a**–**f**) are the variation process in an interval of four hours.

Shown in Figure 12 are coseismic ionospheric distributions in the Tohoku-Oki earthquake, from which we can find that the VTEC increased visibly at 4:00 UTC by comparing the result in Figure 12d (the subfigure of Figure 12a) with those in Figure 12e,f (the subfigure of Figure 12b,c). A similar phenomenon can also be discovered in other researches (e.g., Cai et al. [59]).

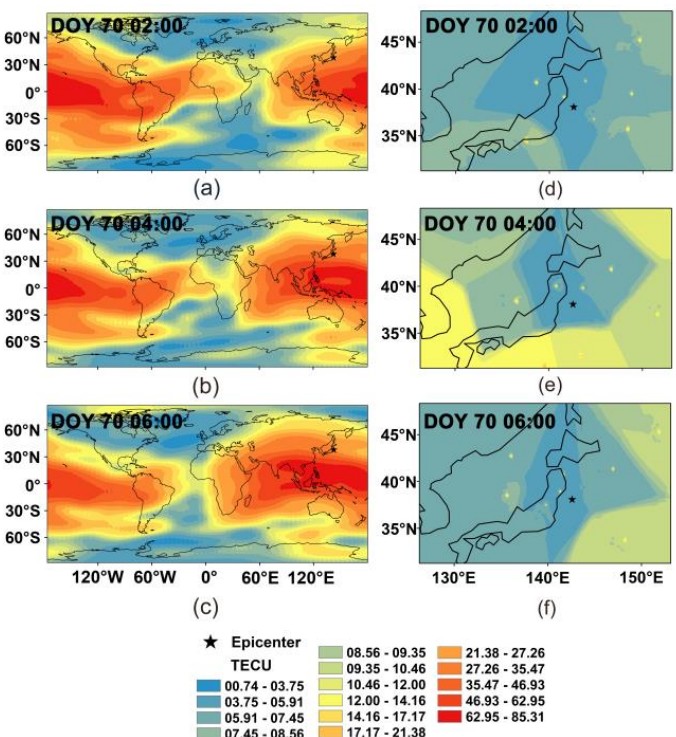

**Figure 12.** Distribution of VTEC near the earthquake epicenter from 2:00 UTC to 6:00 UTC on DOY 70, 2011 (the day earthquake happened); subfigures (**a–c**) are the variation process of GIM data; subfigures (**d–f**) are the variation process of SF-PPP.

## 4. Conclusions

In this paper, a SF-PPP model based on ionospheric delay and receiver DCB constrained is presented to invert the coseismic deformation and the ionospheric disturbance before and after an earthquake. The GPS data from the 2013 Lushan and 2011 Tohoku-Oki earthquakes were analyzed to assess the proposed method.

According to the results, (1) the proposed SF-PPP method provided coseismic deformation signals correctly, although its positioning precision was lower than GAMIT/TRACK. (2) On the other hand, compared to GAMIT/TRACK, the proposed SF-PPP was not affected by the selection of the reference station. Thus, it was not degraded by the common mode errors that existed in the GAMIT/TRACK solution, especially during a strong earthquake (i.e., the Tohoku-Oki one). (3) While analyzing the estimated VTEC, it was found that negative VTEC anomalies appeared visibly around the epicenter of Lushan earthquake on the long-term scale, comparing the results in DOY 105 to those in DOY 106. In contrast, on the short-term scale, negative VTEC anomalies appeared during the several hours before and after the earthquake. For the VTEC obtained from GIM data, negative VTEC anomalies appeared visibility in the south area of the epicenter on the day the earthquake occurred. The VTEC calculated by SF-PPP adopted more observation stations in Sichuan; thus, it is possible to present the ionospheric variations in a smaller region and on a shorter time scale. However, the observations of GIM data are sparser than those of SF-PPP. Thus, the variation of VTEC is more obvious in larger regions (e.g., the whole world).

The presented method has been proved effective for obtaining the deformation and ionospheric disturbance before, during, and after earthquakes. However, it might be

challenging to use such a method for accurate natural-hazard monitoring. Currently, there is rapid development of multi-sensor fusion technologies (i.e., GNSS, seismograph, inertial sensors, crack meters, etc.) Meanwhile, the Chinese government is paying more attention to the development of the multi-source data fusion algorithm for contingent nature hazards monitoring and predicting. Thus, our group will carry out more work in those fields in the future.

**Author Contributions:** Conceptualization, J.L. and Z.G.; data curation, J.L. and Y.W.; funding acquisition, C.Y. and J.P.; investigation, J.L. and Z.G.; software, Z.G.; visualization, C.Y. and J.P.; writing—original draft preparation, J.L.; writing—review and editing, J.L. and Z.G. All authors have read and agreed to the published version of the manuscript.

**Funding:** This research was partly supported by the National Key Research and Development Program of China (Grant No. 2020YFB0505802), and the National Natural Science Foundation of China (NSFC) (Grants No. 42074004).

**Data Availability Statement:** The datasets adopted in this paper are managed by the School of Land Science and Technology, China University of Geosciences, Beijing, and can be available on request from the corresponding author.

**Acknowledgments:** The authors would like to thank anonymous reviewers who gave valuable suggestion that has helped to improve the quality of the manuscripts.

**Conflicts of Interest:** The authors declare no conflict of interest.

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
