# Peer review of "Investigation of Displacement and Ionospheric Disturbance during an Earthquake Using Single-Frequency PPP"

_remotesensing, doi:10.3390/rs14174286_

Round 1
Reviewer 1 Report
In the manuscript the authors present an undifferenced uncombined Single frequency GNSS precise point positioning (SF-PPP) approach for earthquake-displacement estimation. While estimating the GNSS receiver displacement, the proposed method takes into account the contribution of the ionosphere and the DCBs with separate parameters. Case studies are carried out with the method around two earthquakes, where the method is compared with the more established TRACK module of GAMIT/GLOBK.
The article is well organised and contains extensive references to previous studies. Nevertheless, some major points should be clarified before the possible publication.
Major comments:
I) The manuscript highlights the usability of the method during earthquakes as its main contribution. This is of course a great merit, but since the method itself contains new steps that have not been verified and validated before, shouldn't the main point be the method itself? Now it seems that the differences and effects between different high-precision positioning methods and single- and dual-frequency methods are mixed in a complex coseismic situation.
Therefore, to me it would seem more logical to start with a validation in a more controlled situation where verified measurements of the ionospheric contribution and DCBs produced by other means are available.
In addition, the differences between the Post processing kinematic (PPK) and Precise Point Positioning (PPP) in its more standard dual-frequency version are somewhat known and listed in the introduction. Thus, instead of TRACK module of GAMIT/GLOBK, which is based on PPK, wouldn't a comparison with the dual-frequency PPP method would seem more appropriate?
II) In general, it is difficult to follow the methodology on the basis of the manuscript alone. The methodology section needs focusing on the specific method used in the study.
My main concern here is in section 2.3. which is written in a very general form without proper connection to previous or following sections.
The GNSS observables in 2.1 are introduced as undifferenced uncombined pseudorange and carrier phase. But these are also the basic GNSS observables. It would be clearer to define the "un" negations only after mentioning what the combined and differenced versions mean.
Identical or near-identical versions of pseudorange and carrier phase equations are presented several times.
Minor comments:
1. In 2.4 you mention reference stations and monitoring stations. Does the monitoring station refer to individual arbitrary pixels where the VTEC is interpolated?
2. Reference to densest GPS networks in the world.
3. Lines 298 and 299: How was the selection made?
4. Line 320: Is this the only possible explanation?
5. Line 445: Why just these days?
6. The layout in Figures 8 and 11 makes the comparison difficult. How about grouping similarly to Figure 10?
7. Figure 11: Please connect the subplots and times explicitly.
Author Response
Review response 1 for manuscript remotesensing-1870641
Reviewer #1:
In the manuscript the authors present an undifferenced uncombined Single frequency GNSS precise point positioning (SF-PPP) approach for earthquake-displacement estimation. While estimating the GNSS receiver displacement, the proposed method takes into account the contribution of the ionosphere and the DCBs with separate parameters. Case studies are carried out with the method around two earthquakes, where the method is compared with the more established TRACK module of GAMIT/GLOBK.
The article is well organised and contains extensive references to previous studies. Nevertheless, some major points should be clarified before the possible publication.
Major comments:
- I) The manuscript highlights the usability of the method during earthquakes as its main contribution. This is of course a great merit, but since the method itself contains new steps that have not been verified and validated before, shouldn't the main point be the method itself? Now it seems that the differences and effects between different high-precision positioning methods and single- and dual-frequency methods are mixed in a complex coseismic situation.
Therefore, to me it would seem more logical to start with a validation in a more controlled situation where verified measurements of the ionospheric contribution and DCBs produced by other means are available.
In addition, the differences between the Post processing kinematic (PPK) and Precise Point Positioning (PPP) in its more standard dual-frequency version are somewhat known and listed in the introduction. Thus, instead of TRACK module of GAMIT/GLOBK, which is based on PPK, wouldn't a comparison with the dual-frequency PPP method would seem more appropriate?
Answer: In this paper, the purpose is to present the potential possibility of using the ionospheric delay and receiver DCB constrained single-frequency PPP to investigate the displacement and ionospheric disturbance during an earth-quake. Therefore, the high-accuracy results of the dual-frequency TRACK are used as an authoritative comparison, and no results of dual-frequency PPP are provided. As we know, there is no other software which can provide a reliable result based on single-frequency observations at present. Therefore, we choose the dual-frequency based results from the TRACK as the reference comparison. The advantage of the TRACK-based result is of high accuracy. But its disadvantage is easily affected by common mode error like in the case of Tohoku-Oki earthquake, which just highlights the advantage of our method. In addition, if we compare the TRACK-based result with that of dual-frequency PPP, it will completely derive from the original purpose of this paper. The dual-frequency PPP based earthquake works have been studied widely by many scholars.
- II) In general, it is difficult to follow the methodology on the basis of the manuscript alone. The methodology section needs focusing on the specific method used in the study.
My main concern here is in section 2.3. which is written in a very general form without proper connection to previous or following sections.
Answer: Section 2.3 is the algorithm of parameter vector estimation. To make it present connection with the previous section, we have added the corresponding parameter vector description in section 2.2, which can be found in the revised paper.
The GNSS observables in 2.1 are introduced as undifferenced uncombined pseudorange and carrier phase. But these are also the basic GNSS observables. It would be clearer to define the "un" negations only after mentioning what the combined and differenced versions mean.
Answer: The “undifferenced uncombined pseudorange and carrier phase” is a relative description to these ionospheric-free combination (i.e., in Eqs (8) and (9)) and the differenced combination (i.e., dual-differenced observations in PPK), which is a common description in the PPP-related fields (i.e., in Li et al. (2013) and Tu et al. (2013)). Therefore, the corresponding expression is also used in this part.
Identical or near-identical versions of pseudorange and carrier phase equations are presented several times.
Answer: We have deleted the unnecessary repeated equations in the revised paper.
Minor comments:
- In 2.4 you mention reference stations and monitoring stations. Does the monitoring station refer to individual arbitrary pixels where the VTEC is interpolated?
Answer: Here, the monitoring stations refer to the IPP point where the VTEC value is estimated by SF-PPP related to the reference stations, and we modified the corresponding expressions in the revised paper.
- Reference to densest GPS networks in the world.
Answer: We have corrected the description in the revised paper.
- Lines 298 and 299: How was the selection made?
Answer: We chose the stations according to other engaged researches (i.e., in Song et al. (2017)) and the distance between the GPS stations and the epicenter, and we make it clear in the revised paper.
- Line 320: Is this the only possible explanation?
Answer: We checked it and according to engaged researches (i.e., in Song et al. (2017)) this is the most possible explanation for this phenomenon.
- Line 445: Why just these days?
Answer: According to the result of Jiang et al. (2013) the solar activity was at a low level except these days. The details can be found in their work.
- The layout in Figures 8 and 11 makes the comparison difficult. How about grouping similarly to Figure 10?
Answer: Figures 8 and 11 are different results in different time scales to illustrate the variation of regional VTEC. So, it is difficult to be grouped.
- Figure 11: Please connect the subplots and times explicitly.
Answer: We have modified the subplots in the revised paper.
Jiang, W.; Ma, Y.; Liu, H.; Deng, L.; Zhou, X. Investigation of Lushan earthquake ionosphere VTEC anomalies based on GPS data. Earthquake Science. 2013, 26(3-4), 259-265.
Li, X.; Ge, M.; Zhang, H.; Wickert, J. A method for improving uncalibrated phase delay estimation and ambiguity-fixing in real-time precise point positioning. Journal of Geodesy. 2013, 87(5), 405-416.
Tu, R.; Ge, M.; Zhang, H.; Huang, G. The realization and convergence analysis of combined PPP based on raw observation. Advances in space research. 2013, 52(1), 211-221.
Song, C.; Xu, C.; Wen, Y.; Yi, L.; Xu W. Surface deformation and early warning magnitude of 2016 Kaikoura (New Zealand) earthquake from high-rate GPS observations. Chinese Journal of Geophysics. 2017, 60(6), 602-612.
Your comments and suggestions are highly appreciated.
Jie Lv, Zhouzheng Gao, Cheng Yang, Yingying Wei and Junhuan Peng.

Reviewer 2 Report
The article is readable, not wordy. It discusses an important topic.
It is definitely original. Nevertheless - the single-frequency preudorange and phase combination is an order of magnitude less accurate than the two-frequency phase combination.
Since it is stated that the phenomenon in question - an earthquake - needs to be monitored by a set of sensors, some of which are quite expensive, it seems to me that saving money on a GNSS receiver is not the best solution - when the price difference of a single-frequency and dual-frequency single-board receiver is negligible.
It is definitely important to explore this option (single-frequency receiver and ionospheric modeling), which makes the article completely. Although the resulting ionospheric disturbances do not come across as very convincing fo me.
A few comments for improvement:
missing IGS citation - line 71 - Johnston, G., Riddell, A., Hausler, G. (2017). The International GNSS Service. Teunissen, Peter J.G., & Montenbruck, O. (Eds.), Springer Handbook of Global Navigation Satellite Systems (1st ed., pp. 967-982). Cham, Switzerland: Springer International Publishing. DOI: 10.1007/978-3-319-42928-1
line 141 and following - equations (1) to (3) are based on something - again there is no citation here - there are many sources.
line 215: "...is the DCB is the time..." - incomprehensible
line 353: "...near the field ..." - shouldn't it be near the epicenter?
line 440: the link is wrong, it should be http://wds.kugi.kyoto-u.ac.jp/
Around the line 46 could also be cited the article:
Stankova, H; Kostelecky, J and Novosad, M: An Innovative Approach to Accuracy of Co-Seismic Surface Displacement Detection Using Satellite GNSS Technology. Applied Sciences. 2021, 11(6), 2800, doi: 10.3390/app11062800
It would help the publisher with whom you are preparing the publication.
Author Response
Review response 2 for manuscript remotesensing-1870641
Reviewer #2:
The article is readable, not wordy. It discusses an important topic.
It is definitely original. Nevertheless - the single-frequency preudorange and phase combination is an order of magnitude less accurate than the two-frequency phase combination.
Since it is stated that the phenomenon in question - an earthquake - needs to be monitored by a set of sensors, some of which are quite expensive, it seems to me that saving money on a GNSS receiver is not the best solution - when the price difference of a single-frequency and dual-frequency single-board receiver is negligible.
It is definitely important to explore this option (single-frequency receiver and ionospheric modeling), which makes the article completely. Although the resulting ionospheric disturbances do not come across as very convincing fo me.
A few comments for improvement:
- missing IGS citation - line 71 - Johnston, G., Riddell, A., Hausler, G. (2017). The International GNSS Service. Teunissen, Peter J.G., & Montenbruck, O. (Eds.), Springer Handbook of Global Navigation Satellite Systems (1st ed., pp. 967-982). Cham, Switzerland: Springer International Publishing. DOI: 10.1007/978-3-319-42928-1
Answer: We have added the reference in the revised paper.
- line 141 and following - equations (1) to (3) are based on something - again there is no citation here - there are many sources.
Answer: We have added the corresponding citations in this part in the revised paper.
- line 215: "...is the DCB is the time..." - incomprehensible
Answer: We have modified the expressions in the revised paper.
- line 353: "...near the field ..." - shouldn't it be near the epicenter?
Answer: We have corrected the description in the revised paper.
- line 440: the link is wrong, it should be http://wds.kugi.kyoto-u.ac.jp/
Answer: We have corrected the website in the revised paper.
- Around the line 46 could also be cited the article:
Stankova, H; Kostelecky, J and Novosad, M: An Innovative Approach to Accuracy of Co-Seismic Surface Displacement Detection Using Satellite GNSS Technology. Applied Sciences. 2021, 11(6), 2800, doi: 10.3390/app11062800
Answer: We have added the reference in the revised paper.
Your comments and suggestions are highly appreciated.
Jie Lv, Zhouzheng Gao, Cheng Yang, Yingying Wei and Junhuan Peng.

Reviewer 3 Report
This paper introduces a single-frequency PPP model based on ionospheric constraint and receiver DCB to investigate earthquake deformation and VTEC disturbance with two earthquake cases. In general, the work is acceptable because it provides a potential thought in earthquake investigation. But I have some questions the authors should further explain or correct.
1. Figure 6. there is an important motion especially for the station SCXJ on the vertical direction. The authors should explain the reason.
2. I think the repeated descriptions in the figure captions should be simplified (e.g. 8:00 UTC and 10:00 UTC in each figures).
3. The description should keep consistent (e.g. Vertical Total Electron Content (VTEC) in abstract and keywords).
4. The reference form should be further revised.
The authors should check the full manuscript carefully, then correct all the inappropriate descriptions.
Author Response
Review response 3 for manuscript remotesensing-1870641
Reviewer #3:
This paper introduces a single-frequency PPP model based on ionospheric constraint and receiver DCB to investigate earthquake deformation and VTEC disturbance with two earthquake cases. In general, the work is acceptable because it provides a potential thought in earthquake investigation. But I have some questions the authors should further explain or correct.
- Figure 6. there is an important motion especially for the station SCXJ on the vertical direction. The authors should explain the reason.
Answer: We have considered the motion of the vertical component earnestly and checked it from some published works.
Because of the worse precision of vertical component of GPS positioning and week displacement signals in vertical, seismic signals are easy to be submerged by noise that makes it difficult to be detected. Besides, according to Du et al. (2013), there are slips that occurred on day 110 at both SCTQ and SCXJ, which makes it difficult to distinguish the vertical displacements.
- I think the repeated descriptions in the figure captions should be simplified (e.g. 8:00 UTC and 10:00 UTC in each figures).
Answer: We have modified the corresponding expressions in the revised paper.
- The description should keep consistent (e.g. Vertical Total Electron Content (VTEC) in abstract and keywords).
Answer: We have modified all the expressions in the manuscript in the revised paper.
- The reference form should be further revised.
Answer: We have modified the references in the revised paper.
The authors should check the full manuscript carefully, then correct all the inappropriate descriptions.
Answer: We have checked and corrected all the inaccurate expressions in the revised paper.
Du, Y.; Wang, Z.; Yang, S.; An, J.; Liu, Q.; Che, G. Co-seismic deformation derived from GPS observations during April 20th, 2013 Lushan Earthquake, Sichuan, China. Earthquake Science. 2013, 26(3-4), 153-160.
Your comments and suggestions are highly appreciated.
Jie Lv, Zhouzheng Gao, Cheng Yang, Yingying Wei and Junhuan Peng.

Round 2
Reviewer 1 Report
The authors have responded my questions and comments and made modifications correspondingly. However, there are still some points that are so unclear to me that I would need to have more detailed answers before I could recommend publication.
It may not be necessary to make major changes to the manuscript if the authors are able to answer my questions exhaustively, which may well reflect my own ignorance of this particular application.
1. In relation to the first comment of the previous review: I understand that you are interested specifically in the earthquake situation, but don't you think the presented SF-PPP should be verified/validated first in a better controlled situation? If the method were used during seismically quiet times, it should be able to estimate both the ionospheric contribution and the DCBs and it could be validated with actual ionospheric measurements and with independently estimated DCB's? Have you carried out any comparisons regarding the DCBs in the current situation?
2. Previous continued: I am not saying you should compare TRACK to regular dual-frequency PPP. What I am saying is that to my eyes the conducted research would seem significantly stronger if the validation of SF-PPP would be against dual-frequency PPP. On the other hand, I understand that the validation by a completely different method is too laborious at this stage. However, I think you should discuss why you chose what seems a less obvious reference point, or kindly tell me why this is not relevant.
3. Please be still more explicit with the details in the Sections 2.2 and 2.3 or add a reference that is. Is the weight matrix P_k the measurement error covariance inverse or where do the weights come from? How is the initial prior covariance matrix \Sigma_{X0} formed? Related to this, what is the initial value for DCBs and how strictly it is limited by its prior variance?
If the notations separated the variables and their estimators, it would be easier to follow the formulae.
4. The sequential least square estimation method doesn't consist of dynamical step (as a Kalman filter would). So in practice the system solves for static parameters as while solving for different time steps the measurements are weighted only by their measurement errors (i.e. if these weights would be the same, oldest and latest measurements would have the same weight in the solution). Is this correct? Is the analysis run sequentially for the whole interval, or in some larger batches? Again, a better reference would probably explain this.
Author Response
Review response 1 for manuscript remotesensing-1870641
Reviewer #1:
The authors have responded my questions and comments and made modifications correspondingly. However, there are still some points that are so unclear to me that I would need to have more detailed answers before I could recommend publication.
It may not be necessary to make major changes to the manuscript if the authors are able to answer my questions exhaustively, which may well reflect my own ignorance of this particular application.
Point 1: In relation to the first comment of the previous review: I understand that you are interested specifically in the earthquake situation, but don't you think the presented SF-PPP should be verified/validated first in a better controlled situation? If the method were used during seismically quiet times, it should be able to estimate both the ionospheric contribution and the DCBs and it could be validated with actual ionospheric measurements and with independently estimated DCB's? Have you carried out any comparisons regarding the DCBs in the current situation?
Response 1: Many thanks for your comments, actually we have done some work on these points. For example, the accuracy of the estimated ionospheric delay and receiver DCB has been evaluated in our only works (Zhang et al 2013), also the influence of with and without receiver DCB estimation has been evaluated in this work. Recently, our previous work (Lv et al. 2021) validated the positioning performance, ionospheric delay, and receiver DCB of GPS+BDS-2 under kinematic single-/dual-/triple-frequency PPP, and Gao et al. 2017 assessed the performance of different SF-PPP models and got the conclusion that the ionospheric delay and receiver DCB constrained SF-PPP is the best one.
Zhang, H.; Gao, Z.; Ge, M.; Niu, X.; Huang, L.; Tu, R.; Li, X. On the convergence of ionospheric constrained precise point positioning (IC-PPP) based on undifferential uncombined raw GNSS observations. Sensors. 2013, 13(11), 15708–15725.
Lv, J.; Gao, Z.; Kan, J.; Lan, R.; Li, Y.; Lou, Y.; Yang, H.; Peng, J. Modeling and assessment of multi-frequency GPS/BDS-2/BDS-3 kinematic precise point positioning based on vehicle-borne data. Measurement. 2022, 189, 110453.
Gao, Z.; Ge, M.; Shen, W.; Zhang, H.; Niu, X. Ionospheric and receiver DCB-constrained multi-GNSS single-frequency PPP integrated with MEMS inertial measurements. Journal of Geodesy. 2017, 91(11), 1351-1366.
Point 2: Previous continued: I am not saying you should compare TRACK to regular dual-frequency PPP. What I am saying is that to my eyes the conducted research would seem significantly stronger if the validation of SF-PPP would be against dual-frequency PPP. On the other hand, I understand that the validation by a completely different method is too laborious at this stage. However, I think you should discuss why you chose what seems a less obvious reference point, or kindly tell me why this is not relevant.
Response 2: SORRY for the last answer we did not get your real point. Yes, as we know, actually the dual-frequency PPP has been applied in such fields. We provided some of these nice works in the introduction part. Since the previous works are mainly based on the dual-frequency PPP, and few works are based on single-frequency PPP, especially the ionospheric delay and receiver DCB constrained SF-PPP. As we have done some works (Zhang et al 2013; Lv et al. 2021; Gao et al. 2017) on such SF-PPP model, therefore, we want to know how about using such SF-PPP model during an earthquake. Then, we did the corresponding work in this paper and did not provide the solutions for dual-frequency PPP.
Zhang, H.; Gao, Z.; Ge, M.; Niu, X.; Huang, L.; Tu, R.; Li, X. On the convergence of ionospheric constrained precise point positioning (IC-PPP) based on undifferential uncombined raw GNSS observations. Sensors. 2013, 13(11), 15708–15725.
Lv, J.; Gao, Z.; Kan, J.; Lan, R.; Li, Y.; Lou, Y.; Yang, H.; Peng, J. Modeling and assessment of multi-frequency GPS/BDS-2/BDS-3 kinematic precise point positioning based on vehicle-borne data. Measurement. 2022, 189, 110453.
Gao, Z.; Ge, M.; Shen, W.; Zhang, H.; Niu, X. Ionospheric and receiver DCB-constrained multi-GNSS single-frequency PPP integrated with MEMS inertial measurements. Journal of Geodesy. 2017, 91(11), 1351-1366.
Point 3: Please be still more explicit with the details in the Sections 2.2 and 2.3 or add a reference that is. Is the weight matrix P_k the measurement error covariance inverse or where do the weights come from? How is the initial prior covariance matrix \Sigma_{X0} formed? Related to this, what is the initial value for DCBs and how strictly it is limited by its prior variance?
If the notations separated the variables and their estimators, it would be easier to follow the formulae.
Response 3: We have added a reference in the revised paper. The conventional satellite elevation-dependent weight function (Gendt et al. 2003) to calculate the prior variance.
where and are the elevation of satellites and the corresponding prior variance.
The initial prior covariance of code and phase (Teunissen and Montenbruck, 2017) is afforded by conventional values.
The initial DCB prior covariance we adopted here is 0.3.
Gendt, G.; Dick, G.; Reigber, C.H.; Tomassini, M.; Liu, Y. Demonstration of NRT GPS water vapor monitoring for numerical weather prediction in Germany. Journal of the Meteorological Society of Japan. 2003, 82(1B), 360–370.
Teunissen, P. J.; Montenbruck, O. (Eds.) Springer handbook of global navigation satellite systems (Vol. 10, pp. 978-3). 2017, New York, NY, USA: Springer International Publishing.
Point 4: The sequential least square estimation method doesn't consist of dynamical step (as a Kalman filter would). So in practice the system solves for static parameters as while solving for different time steps the measurements are weighted only by their measurement errors (i.e. if these weights would be the same, oldest and latest measurements would have the same weight in the solution). Is this correct? Is the analysis run sequentially for the whole interval, or in some larger batches? Again, a better reference would probably explain this.
Response 4: As described in Fu et al. 2018, the sequential least square estimation can be adopted in GNSS precise data processing. However, as you mentioned, it doesn't present the state-update step, so for static parameters, the continuous measurements without cycle slip are weighted only by their measurement errors (i.e., the satellite elevation-dependent weight function, Gendt et al. 2003). The analysis runs sequentially according to GNSS observation epochs here (1Hz in this paper).
Gendt, G.; Dick, G.; Reigber, C.H.; Tomassini, M.; Liu, Y. Demonstration of NRT GPS water vapor monitoring for numerical weather prediction in Germany. Journal of the Meteorological Society of Japan. 2003, 82(1B), 360–370.
Fu, W.; Yang, Y.; Zhang, Q.; Huang, G. Real-time estimation of BDS/GPS high-rate satellite clock offsets using sequential least squares. Advances in Space Research. 2018, 62(2), 477-487.
Many thanks again for your helpful suggestions and comments, they are highly appreciated.
Jie Lv, Zhouzheng Gao, Cheng Yang, Yingying Wei and Junhuan Peng.
